# Acidic CO₂-to-HCOOH electrolysis with industrial-level current on phase engineered tin sulfide

Haifeng Shen [1,3], Huanyu Jin [1,3], Haobo Li [1,3], Herui Wang[2], Jingjing Duan [2], Yan Jiao [1] & Shi-Zhang Qiao [1] ✉

Acidic CO₂-to-HCOOH electrolysis represents a sustainable route for value-added CO₂ transformations. However, competing hydrogen evolution reaction (HER) in acid remains a great challenge for selective CO₂-to-HCOOH production, especially in industrial-level current densities. Main group metal sulfides derived S-doped metals have demonstrated enhanced CO₂-to-HCOOH selectivity in alkaline and neutral media by suppressing HER and tuning CO₂ reduction intermediates. Yet stabilizing these derived sulfur dopants on metal surfaces at large reductive potentials for industrial-level HCOOH production is still challenging in acidic medium. Herein, we report a phase-engineered tin sulfide pre-catalyst (π-SnS) with uniform rhombic dodecahedron structure that can derive metallic Sn catalyst with stabilized sulfur dopants for selective acidic CO₂-to-HCOOH electrolysis at industrial-level current densities. In situ characterizations and theoretical calculations reveal the π-SnS has stronger intrinsic Sn-S binding strength than the conventional phase, facilitating the stabilization of residual sulfur species in the Sn subsurface. These dopants effectively modulate the CO₂RR intermediates coverage in acidic medium by enhancing *OCHO intermediate adsorption and weakening *H binding. As a result, the derived catalyst (Sn(S)-H) demonstrates significantly high Faradaic efficiency (92.15 %) and carbon efficiency (36.43 %) to HCOOH at industrial current densities (up to −1 A cm⁻²) in acidic medium.

Electrochemical CO₂ reduction reaction (CO₂RR) to formic acid (HCOOH) provides a promising strategy for renewable energy storage and CO₂ recycling[1–3]. Specifically, HCOOH is an ideal liquid intermediate for hydrogen storage with high energy density[4,5]. Previous studies mainly focus on suppressing hydrogen evolution reaction (HER) in alkaline or near-neutral solutions to improve selectivity. However, these processes exhibit disadvantages for practical applications: (1) Formate (HCOO⁻) instead of HCOOH is obtained in alkaline or near-neutral electrolytes because the dissociation constant (pKa) of HCOOH is 3.75[6]. Therefore, additional conversion and separation treatment are required to obtain HCOOH. (2) Over 75% of input CO₂ has been consumed by reacting with hydroxide (OH⁻), and the dominant by-products carbonate (CO₃²⁻) or bicarbonate (HCO₃⁻) together with HCOO⁻ would pass anion exchange membrane (AEM) to the anode and then be re-oxidized to CO₂, resulting in low carbon efficiency and product loss[7–9]. (3) Continuous input of CO₂ reduces the pH value of the electrolyte, leading to the poor stability of the catalysts and membrane. Therefore, electrolyte needs to be renewed routinely, especially in alkaline medium[10–12].

Alternatively, acidic CO₂-to-HCOOH electrolysis can solve the above problems well. In acidic solution (pH < 3.75), CO₃²⁻ or HCO₃⁻ by-product formation was inhibited, so HCOOH is produced directly and

[1]School of Chemical Engineering, The University of Adelaide, Adelaide, SA 5005, Australia. [2]School of Energy and Power Engineering, Nanjing University of Science and Technology, 210094 Nanjing, China. [3]These authors contributed equally: Haifeng Shen, Huanyu Jin, Haobo Li. ✉e-mail: s.qiao@adelaide.edu.au

easily extracted and purified. Furthermore, using proton exchange membrane (PEM) in acidic $CO_2RR$ prevents the migration of HCOOH to the anode[7,13], which avoids product loss. Therefore, the acidic medium is more suitable for $CO_2$ protonation to HCOOH than alkaline and neutral media[14]. However, the high hydronium ion concentration in acid substantially promotes HER, engendering lower Faradaic efficiency (FE) of $CO_2RR$ products[6,15]. Hence, suppressing HER in the acidic $CO_2RR$ is imperative yet challenging.

Recent studies of acidic $CO_2RR$ mainly focus on alkali-cation strategy, which significantly suppresses HER and promotes $CO_2RR$ in acidic medium by tuning the electric field in the double layer to inhibit the migration of hydronium ions toward cathode[6,16–18]. For example, Gu et al. reported the $SnO_2$/C pre-catalyst with 0.8 M $K^+$ ions for efficient $CO_2$-to-HCOOH production in acid media, which illustrated a $-220$ mA $cm^{-2}$ of maximum HCOOH partial current density and the FE up to 88 %, respectively[6]. Qiao et al. proposed high concentration (3 M) of $K^+$-assisted $CO_2RR$ with Bi nanosheets in an acidic medium to produce HCOOH, as FE reached 92.2% with the current density of $-237.1$ mA $cm^{-2}$, and achieved carbon efficiency of 27.4 %[16]. However, ampere-level current of $CO_2$-to-HCOOH production in acidic medium has not been achieved using this strategy because HER gradually dominates the overall reaction when the applied current densities are over $-300$ mA $cm^{-2}$. Thus, acidic $CO_2$-to-HCOOH electrolysis at industrial current density (over $-300$ mA $cm^{-2}$) still remains a great challenge. Considering the higher $CO_2$ concentration in acidic medium[19–21], we reasoned that modulating the coverage of $CO_2RR$ reaction intermediates is a possible way to suppress HER and promote $CO_2$-to-HCOOH production. Specifically, enhancing the adsorption of *OCHO on the catalyst interfaces and weakening the adsorption of *H or $H_2O$ have the potential to optimize the $CO_2RR$ intermediates coverage under high current density[7]. Therefore, selective HCOOH production may be effectively improved in this way.

Main group metal sulfides (SnS, $In_2S_3$, etc.) were widely employed as the pre-catalysts for $CO_2$-to-HCOOH production because the derived S-doped main group metal catalysts show enhanced selectivity for HCOOH/HCOO$^-$ formation with inhibition of HER[22–24]. Practically, the S dopants in the main group metals can optimize the adsorption of $CO_2RR$ intermediates and suppress the hydrogen or water adsorption. For example, Ma et al. reported S-doped indium (S-In) catalyst for enhanced HCOO$^-$ production in near-neutral media with 93% of FE and $-84$ mA $cm^{-2}$ partial current density, which is better than In, Se-In and Te-In catalysts. Therefore, tailoring S-doping is an efficient way to promote HCOO$^-$ production in alkaline or near-neutral media at current densities below $-200$ mA $cm^{-2}$. However, the industrial-level current of $CO_2$-to-HCOOH electrolysis in acidic medium was still significantly limited due to the over-reduction of S dopants at high overpotentials, followed by a rapid increase in HER. Therefore, stabilizing S dopants on metals subsurface at industrial current densities is crucial to promote acidic $CO_2$-to-HCOOH production.

In this study, we realized acidic $CO_2$-to-HCOOH production (pH = 3) at industrial current densities by phase engineering of SnS (π-SnS) pre-catalyst. Compared to conventional phase of α-SnS, π-SnS has

stronger metal-S binding strength, leading to higher S residual at large $CO_2RR$ current densities, confirmed by theoretical simulations and in situ characterizations. The high S content in Sn subsurface modulates the *OCHO coverage and promotes $CO_2$-to-HCOOH production at industrial current densities. Density functional theory (DFT) calculations revealed that the S-doped Sn has a stronger *OCHO adsorption and weaker *H adsorption than the pure metallic Sn, leading to high coverage of $CO_2$-to-HCOOH intermediates at high current density for selective HCOOH production. As a result, the π-SnS-derived catalyst illustrates a maximum FE of 92.15% toward HCOOH, a partial HCOOH current density of $-730$ mA $cm^{-2}$, and 36.43% of single-pass $CO_2$ utilization in acidic medium. Our research provides new insight into phase-engineered electrocatalysts for value-added commodity chemicals production.

## Results

Figure 1 illustrates the evolution of π-SnS for acidic $CO_2$-to-HCOOH electrolysis. By employing phase engineering strategy, strong intrinsic binding strength between Sn and S is realized in π-SnS, which stabilizes higher S content in the subsurface of Sn metals after $CO_2RR$ activation. As a result, strong *OCHO adsorption and weak *H adsorption achieved on π-SnS-derived catalyst during $CO_2RR$, and the high coverage of *OCHO intermediate on the S-Sn sites could promote acidic $CO_2$-to-HCOOH production and suppress HER at industrial-level current densities.

The π-SnS was prepared by a hydrothermal method (see Methods), which is achieved by mixing $SnCl_2·2H_2O$, cysteine and polyvinylpyrrolidone (PVP). The field-emission scanning electron microscope (FESEM) images (Fig. 2a, b) and X-ray diffraction (XRD) analysis (Fig. 2c) reveal uniform rhombic dodecahedron structure of π-SnS with a particle size of $262 ± 37$ nm (Supplementary Fig. 1). In addition, the determined π phase in cubic crystal system (a, b, c = 11.83 Å) matches well with the simulated XRD pattern of the π-SnS. It is noteworthy that the π phase has only been discovered recently as a new family of materials. Controllable synthesis of π phase-based materials remains greatly challenging, and only a few studies have developed small-scale π phase-based materials synthesis[25,26]. In this work, we provided a simple and efficient method to successfully synthesize π-SnS with uniform rhombic dodecahedron morphology, which is rare among metal sulfides. For comparison, conventional α phase SnS (α-SnS) refers to definite layered structures with an orthorhombic crystal system (Fig. 2c, Supplementary Fig. 2). The chemical compositions of π-SnS and α-SnS were confirmed by the energy-dispersive X-ray spectroscopy (EDS) with an atomic ratio of ~1:1 between Sn and S (Supplementary Fig. 3). The high-angle annular dark-field scanning transmission electron microscope (HAADF-STEM) and corresponding EDS mapping images further confirm the excellent crystalline structure of π-SnS with the homogeneous distribution of Sn and S elements (Fig. 2d–f). The atomic structures of π-SnS together with corresponding fast Fourier transform (FFT) patterns (Fig. 2d, e) confirm the well-defined cubic structure with exposed (110) facet, which is in agreement with the simulated structure (Fig. 2e, g). In

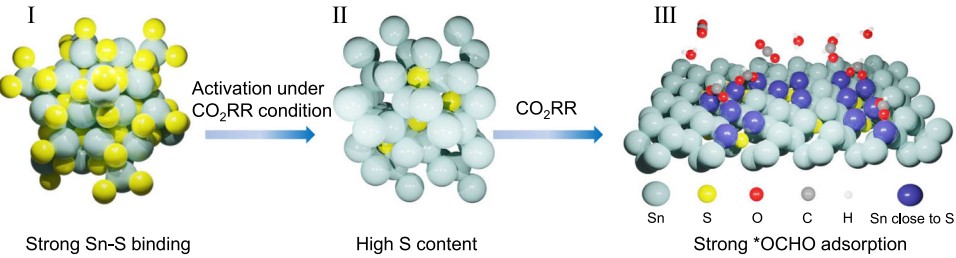

I — Strong Sn-S binding    Activation under $CO_2RR$ condition    II — High S content    $CO_2RR$    III — Strong *OCHO adsorption

Sn    S    O    C    H    Sn close to S

**Fig. 1 | Schematic of phase engineering of SnS pre-catalysts for $CO_2RR$.** I Crystal structure of π-SnS pre-catalyst. II Derived S-doped Sn structure after $CO_2RR$ activation with higher S content than conventional phase. III Strong *OCHO adsorption realized on S-doped Sn surface during $CO_2RR$.

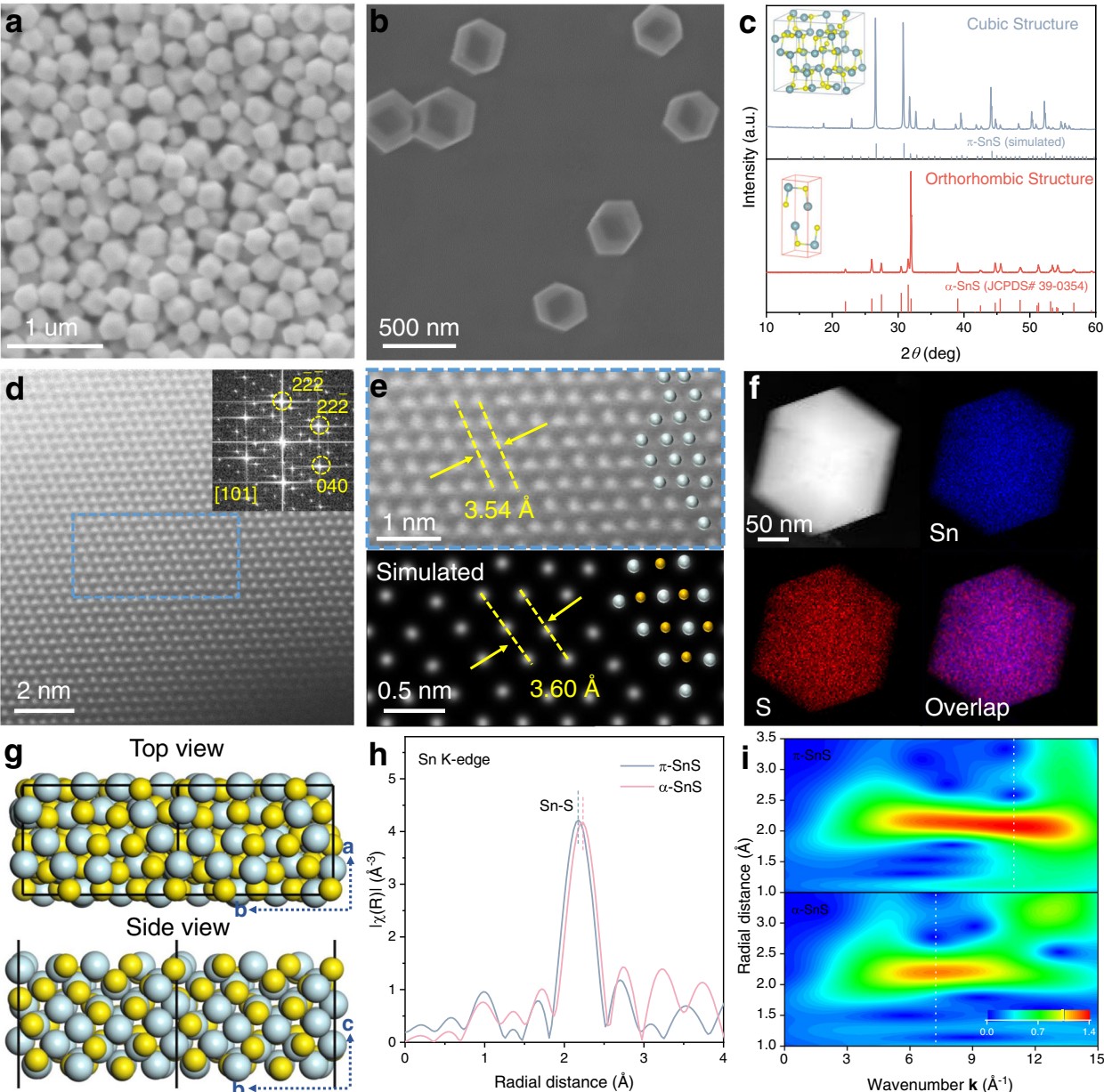

Fig. 2 | Structural characterization of π-SnS pre-catalyst. a,b FESEM images of π-SnS. c XRD patterns of π-SnS (top) and α-SnS (bottom), and the inset of corresponding crystal structures, tin in cyan and sulfur in yellow. d HAADF-STEM image taken in the [101] direction of π-SnS and the inset of FFT pattern. e Magnified HAADF-STEM image taken from the corresponding area in (d) (top), and the simulated microscopic image of π-SnS from [101] direction (bottom). Cyan, tin; yellow, sulfur. f EDS mapping of a π-SnS nanocrystal. g Atomic structures of (110) facet in π-SnS with top view (top) and side view (bottom). h FT-EXAFS spectra for π-SnS and α-SnS at Sn K-edge. i Contour plots of Sn K-edge WT-EXAFS for π-SnS (top) and α-SnS (bottom).

contrast, distinct stacking sequences are observed in α-SnS (Supplementary Fig. 2).

To reveal the different bonding strengths between Sn and S, we further detect the coordination environment in different phases by synchrotron-based X-ray absorption spectroscopy (XAS). The Fourier-transformed extended X-ray absorption fine structure (FT-EXAFS) spectrum of π-SnS illustrates a shorter Sn-S average bond length in the first shell than that in α-SnS (Fig. 2h), which is consistent with the bond length in their model structures (Supplementary Fig. 4). The shorter Sn-S bond in π-SnS means a stronger Sn-S bond energy than α-SnS[27,28]. X-ray absorption near-edge structure (XANES) spectra of Sn K-edge determine the similar oxidation states of Sn in π-SnS and α-SnS (Supplementary Fig. 5), which excludes the effect of high oxidation states

Sn ($Sn^{4+}$) on the binding energy. Wavelet-transform (WT)-EXAFS analyses (Fig. 2i) reveal a distinct larger k-value of π-SnS (11.0 Å$^{-1}$) than that of α-SnS (7.3 Å$^{-1}$). The increased k-values suggest a higher structural order of π-SnS than α-SnS[29,30].

Extensive studies confirm that the reconstitution of various metal compounds at reduction potentials in $CO_2RR$ is a common but complicated phenomenon. Derived catalysts typically display distinct structures from their pre-catalysts, however, this reconstitution process is often ignored. The identification of derived catalysts via reconstitution can contribute to a better understanding of the origin of catalytic performance[31–33]. Here, we investigate the structural differences of the catalysts derived from π-SnS and α-SnS through $CO_2RR$ activation. We performed the $CO_2RR$ reduction for 900 s at

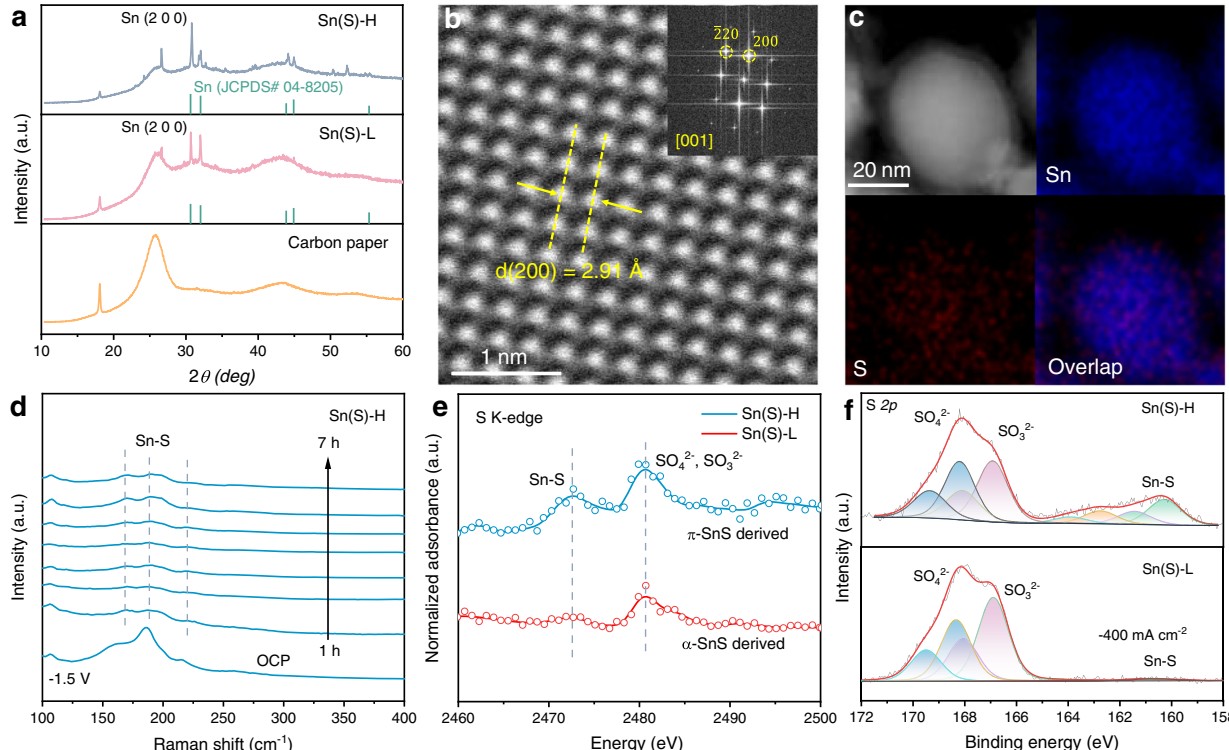

**Fig. 3 | Structural characterization of derived catalysts. a** XRD patterns of derived catalysts on carbon paper after $CO_2RR$ activation. **b,c** Magnified HAADF-STEM image of catalyst derived from π-SnS with the inset of FFT pattern and corresponding EDS elemental mapping. **d** In situ Raman spectra of Sn(S)-H with increasing reduction time at −1.5 V (vs. RHE) in $CO_2$-saturated 0.5 M $K_2SO_4$ solution (pH = 3). **e** S K edge NEXAFS spectra of Sn(S)-H and Sn(S)-L. **f** High-resolution S 2p XPS spectra of Sn(S)-H and Sn(S)-L after $CO_2RR$ activation at −400 mA cm⁻². RHE: reversible hydrogen electrode.

−100 mA cm⁻² to derive S-doped Sn from different phases of SnS (Supplementary Fig. 6). The major differences between the two derived catalysts are the amount of residual S. We observed higher content of S-doping in Sn (Sn(S)-H) and lower content of S-doping in Sn (Sn(S)-L) derived from π-SnS and α-SnS pre-catalysts, respectively. First, metallic Sn peaks in XRD analysis indicate that both π-SnS and α-SnS are reduced to Sn metals with the main exposed (100) facet (Fig. 3a). The weak peak at 26.6 degree should be attributed to the Sn oxidation during XRD test (SnO₂, JCPDS# 70-4177). HAADF-STEM (Fig. 3b) and corresponding elemental mapping images (Fig. 3c, Supplementary Fig. 7) confirm the reconstitution of pre-catalysts with a trace amount of S residual. Furthermore, in situ Raman characterization was performed to trace the structural evolution process of π-SnS and α-SnS pre-catalysts in −1.5 V vs. reversible hydrogen electrode (vs. RHE) (Fig. 3d, Supplementary Fig. 8). The clear Sn-S stretching signal (168 cm⁻¹, 188 cm⁻¹, 220 cm⁻¹) remained well in Sn(S)-H after 7 h reduction, suggesting the good residual content of S element after $CO_2RR$ activation[22,34,35]. However, the Sn(S)-L derived from α-SnS shows undetectable Sn-S stretching only after 3 h reduction, indicating a complete reduction of the material. We further adopted synchrotron-based soft X-ray near-edge X-ray absorption fine structure (NEXAFS) to compare the S content of catalysts derived from π-SnS and α-SnS. The similar broad peaks around 2480.6 eV are attributed to the adsorbed $SO_4^{2-}$ and $SO_3^{2-}$ from the electrolyte, while the retention of S (Sn-S, around 2472.6 eV) in these two samples (Fig. 3e, Supplementary Fig. 8c) are distinctly different. This is because Sn(S)-H has a higher S amount after $CO_2RR$ activation than Sn(S)-L. X-ray photoelectron spectroscopy (XPS) results for the samples derived from π-SnS $CO_2RR$ activation under high current density (−400 mA cm⁻²) display higher S amount, which accords well with NEXAFS data (Fig. 3f). The similar evidence of higher residual S amount in Sn(S)-H could also be found in other sulfur-free acidic electrolytes (Supplementary Fig. 9). Due to the complexity of the reconstitution to

the derived catalysts, we also checked other potential factors such as defects which could affect the catalytic performance[32,33]. As no obvious defects were observed in both Sn(S)-H and Sn(S)-L (Supplementary Fig. 10), we confirm the main difference after reconstitution between Sn(S)-H and Sn(S)-L is the amount of residual S. Such differences of residual S amount in derived catalysts can be attributed to the phase engineering strategy on the pre-catalysts. The stronger binding strength of Sn-S in π-SnS leads to a higher S content in Sn subsurface even under high reductive current density.

Electrochemical $CO_2RR$ measurements for derived Sn(S)-H and Sn(S)-L were conducted in a standard three-electrode flow cell with a gas diffusion electrode (GDE) as the working electrode (see Methods). The pH value of the electrolyte was adjusted to 3.0 using $H_2SO_4$ in 0.5 M $K_2SO_4$ (Supplementary Fig. 11). Linear sweep voltammetry (LSV) curves show both lower onset potentials of Sn(S)-H ( −0.86 V vs. RHE) and Sn(S)-L ( −1.12 V vs. RHE) in $CO_2$-saturated electrolyte than in Ar-saturated electrolyte, indicating a favorable $CO_2RR$ pathway for the two derived catalysts at low overpotentials (Supplementary Fig. 12)[36]. With increasing applied potential, HER on Sn(S)-L gradually dominates, while Sn(S)-H exhibits a better HER suppression in Ar-saturated electrolyte with a larger Tafel slope. In addition, Sn(S)-H has a significantly larger current density in $CO_2$-saturated electrolyte, suggesting a better $CO_2RR$ performance (Supplementary Fig. 12). It is well known that the overall catalytic activity normally consists of two parts: the number of active sites and the intrinsic activity of catalysts[37,38]. The Sn(S)-H catalyst has a lower value of electrochemical surface area (ECSA), confirming a higher intrinsic $CO_2RR$ activity of Sn(S)-H in acidic medium than that of Sn(S)-L (Supplementary Fig. 13).

We further quantified the FE of HCOOH in the derived catalysts in a chronopotentiometry mode with the applied current densities from −0.1 to −1 A cm⁻² (Supplementary Fig. 14). Gas and liquid products were determined using gas chromatography (GC) and ¹H nuclear magnetic resonance (¹H NMR), respectively (see Methods). Sn(S)-H achieved the

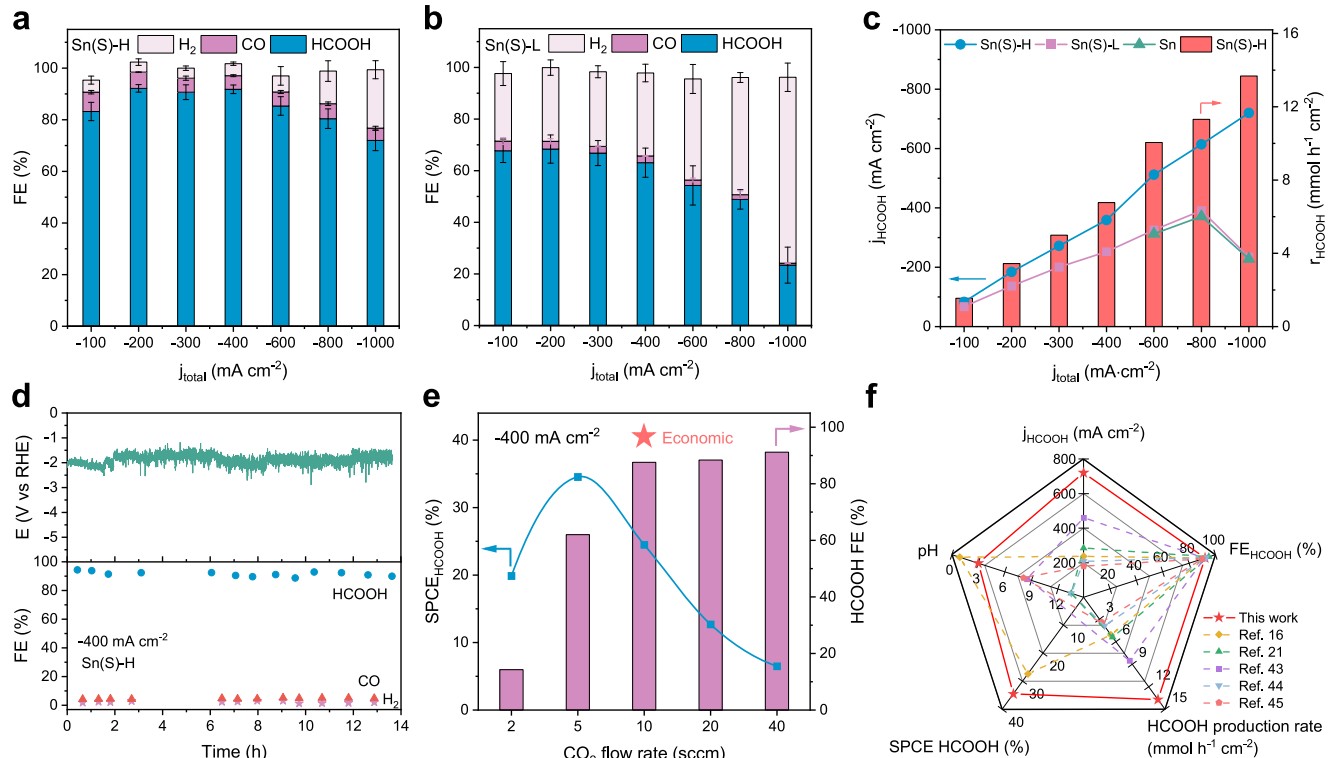

**Fig. 4 | Electrochemical CO₂RR performance in acidic medium. a,b** FE value of CO₂RR products for (**a**) Sn(S)-H and (**b**) Sn(S)-L under different current densities. **c** Partial HCOOH current densities and HCOOH production rates for Sn(S)-H, Sn(S)-L and pure metallic Sn catalysts. **d** Stability measurement of HCOOH production at the total current density of −400 mA cm⁻² for Sn(S)-H. **e** SPCE_HCOOH and FE value of HCOOH for Sn(S)-H at different CO₂ flow rates

under the current density of −400 mA cm⁻². **f** Comparison of this work with other reported catalysts for CO₂RR to produce HCOOH/HCOO⁻, including HCOOH partial current density, FE value of HCOOH, HCOOH production rate, SPCE value of HCOOH and pH of the electrolyte. Error bars correspond to the standard deviation of three independent measurements.

maximum FE of 92.15 ± 1.26% for HCOOH at −200 mA cm⁻², together with over 85% FE for HCOOH in a wide range of current densities (Fig. 4a), corresponding well with ¹H NMR and GC results in each current density (Supplementary Figs. 15–17). In comparison, lower FE (65–70%) of Sn(S)-L was obtained from −100 to −300 mA cm⁻², accompanied by obviously increased HER at high current densities, shown in Fig. 4b. Figure 4c highlights the differences in FE among Sn(S)-H, Sn(S)-L and pure Sn, reflecting a better HER suppression performance of Sn(S)-H. As a result, the Sn(S)-H achieves −730.2 mA cm⁻² partial HCOOH current density and yields a high production rate of 13.7 mmol h⁻¹ cm⁻², which is better than most reported catalysts in alkaline or near-neutral medium (Supplementary Fig. 18, Supplementary Table 1). It is worth noting that pure Sn metal also shows a dominated HER performance at high current densities (Fig. 4c, Supplementary Fig. 19), elucidating the significant role of S dopants in Sn subsurface for efficient acidic CO₂-to-HCOOH production. In addition, when we increased the concentration of hydronium ions at pH = 1, Sn(S)-H maintained good performance in suppressing HER for HCOOH production (Supplementary Fig. 20). For acidic CO₂ reduction, metal dissolution with related long-term stability are important aspects that can not be ignored[32,33,39]. Sn(S)-H shows a 85% FE of HCOOH production under −400 mA cm⁻² in 13.5 h stability measurement (Fig. 4d). In addition, Sn(S)-H maintained metallic Sn structure with stable S-Sn bonding after stability measurement (Supplementary Figs. 21, 22). While Sn(S)-L exhibited continuously decreasing FE of HCOOH for 6 h with a distinctly higher Sn dissolving ratio than Sn(S)-H (Supplementary Fig. 23), which could be attributed to the dissolution of S-Sn sites.

Recently, carbon efficiency of CO₂RR has received constant attention because over 50–75% of input CO₂ is consumed to produce

CO₃²⁻ or HCO₃⁻ in alkaline medium[6,7,40], reducing the economic benefits. Therefore, single-pass carbon efficiency (SPCE) of activated Sn(S)-H catalysts was detected at −400 mA cm⁻² at different CO₂ flow rates (Fig. 4e, Supplementary Fig. 17b). Remarkably, 36.43% of SPEC was achieved (with partial SPCE for HCOOH being 34.54%) during the CO₂ flow rate at 5 standard cc min⁻¹ (sccm). Besides, with the CO₂ flow rate increasing from 2 to 10 sccm at −400 mA cm⁻², the significant growth of FE for HCOOH indicates the positive CO₂RR pathway against HER of Sn(S)-H. Since the economic value of HCOOH is $1000 USD per ton, which far exceeds that of CO₂ of $70 per ton[41,42], it is found that the CO₂-to-HCOOH process at the flow rate of 10 sccm has the highest economic value by balancing carbon efficiency and FE. In addition, the production rate of HCOOH on Sn(S)-H also exceeds most catalysts operated in alkaline/neutral solutions (Fig. 4f)[16,21,43–45].

In situ surface-enhanced Raman spectroscopy (SERS) in flow cell was employed to reveal the reaction mechanism of CO₂-to-HCOOH on Sn(S)-H. No obvious CO₃²⁻ or HCO₃⁻ signal was observed from 1000 to 1100 cm⁻¹, indicating the suppression of carbonate production with a local acid environment on the Sn(S)-H catalyst (Supplementary Fig. 24). The same phenomenon was also observed at high overpotentials from −1.4 to −2 V. The narrow peak at 1550 cm⁻¹ corresponds to asymmetric stretching vibration (ν_as O-C-O) of *OCHO[46,47]. Since no obvious *COOH adsorption peaks were detected, *OCHO was determined as the primary intermediate for acidic CO₂-to-HCOOH in our work, which agrees with the main intermediate of HCOO⁻ formation in neutral and alkaline media[4,48,49].

We further performed in situ attenuated total reflectance Fourier transform infrared spectroscopy (ATR-FTIR) to compare the coverage of *OCHO on Sn(S)-H and Sn(S)-L (Fig. 5a, Supplementary Fig. 25) with the applied potentials from −0.5 to −1.9 V vs. RHE. Similarly, no evident

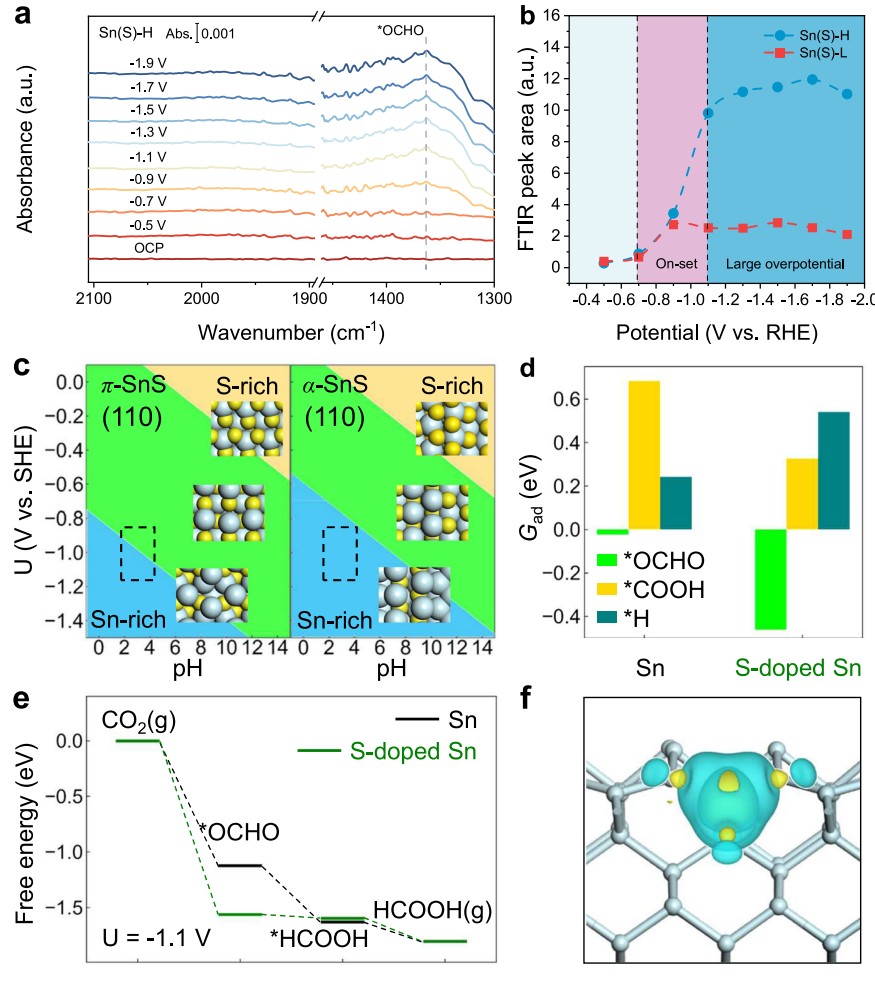

**Fig. 5 | Mechanism investigations. a** In situ ATR-FTIR spectra of Sn(S)-H. **b** FTIR peak areas of Sn(S)-H and Sn(S)-H. **c** Surface Pourbaix diagram of π-SnS (110) and α-SnS (110). The atomic structure (top view) of the S-terminated, stoichiometric and Sn-terminated surface is shown as insets. Cyan sphere: Sn; yellow sphere: S. The working condition of U = −1.1 V (vs. SHE) and pH = 3 is approximately marked in black dashed box. **d** Adsorption free energy ($G_{ad}$) of *OCHO, *COOH and *H on Sn (100) and S-doped Sn (100). **e** Free energy diagram of $CO_2$ electroreduction to HCOOH under U = −1.1 V (vs. SHE) on Sn (100) and S-doped Sn (100). **f** Differential charge density of Sn (100) (side view) with and without S-doping. The yellow and blue contour represents electron accumulation and depletion, respectively. The isosurface level is set to be 0.02 e/Bohr³. The ball-and-stick model in the lower layer shows the positions of atoms.

*CO (to CO product) appeared from 1900 to 2100 cm⁻¹, which agrees with the in situ SERS results. For two $CO_2$-to-HCOOH intermediates, *OCHO is widely considered more efficient than *COOH for HCOOH production[4,36]. As shown in Fig. 5a, the distinct *OCHO signal around 1367 cm⁻¹ on Sn(S)-H together with no obvious *OCHO hydrogenation intermediates detected in in situ ATR-FTIR experiments, confirm the efficient HCOOH generation of Sn(S)-H[34,50]. However, the notable low peak area on Sn(S)-L are mainly due to competitive HER on the catalyst (Supplementary Fig. 25). To further reveal the intrinsic relationship between *OCHO adsorption coverage and $CO_2$RR performance, we compared the FTIR peak areas (around 1342–1380 cm⁻¹) of Sn(S)-H and Sn(S)-L with the function of overpotential (Fig. 5b). Both Sn(S)-H and Sn(S)-L exhibited no evident signal before −0.7 V (vs. RHE) of $CO_2$RR onset. In contrast, the rapid growth of *OCHO peak area from −0.7 to −1.1 V (vs. RHE) on Sn(S)-H can be attributed to the onset of $CO_2$-to-HCOOH production, which is consistent well with the electrochemical data (Supplementary Fig. 12). After that, the increased peak areas in Sn(S)-H on large overpotential region suggest more significant *OCHO coverage than that in Sn(S)-L, which explains the origin of the enhanced $CO_2$-to-HCOOH performance on Sn(S)-H by chronopotentiometry tests. Note that the slightly decreased *OCHO peak

area from −1.7 V to −1.9 V (vs. RHE) suggests the competitive HER in the reaction, which is in accordance with the minor FE loss from −0.4 to −1 A cm⁻².

To understand the formation mechanism of Sn(S)-H and Sn(S)-L catalysts, surface Pourbaix diagrams for the π-SnS and α-SnS were analyzed by density functional theory (DFT) calculations. The (110) facet is chosen for the calculation, corresponding to the dodecahedral morphology of π-SnS pre-catalyst observed in the SEM experiment (Fig. 2b). The most stable surface with the lowest surface free energy under different potential (U) and pH conditions is shown in Fig. 5c. Compared with α-SnS (110), the stoichiometry of π-SnS(110) surface is stable over a wider range of electrochemical conditions (green area). Especially, under the experimental conditions of around U = −1.1 V vs. standard hydrogen electrode (SHE), pH = 3 (black dotted box), α-SnS(110) has completely removed the surface S to exhibit a Sn-rich surface, while π-SnS(110) starts the surface reduction of S. This indicates that the S-atom bonding at the π-SnS surface layer is stronger than that of α-SnS. Therefore, it can be speculated that after the reduction of π-SnS, S atoms are more likely to remain on the surface, corresponding to the higher S atoms content measured experimentally (Fig. 3e, f).

To further reveal the $CO_2RR$ mechanism in derived Sn(S)-H and Sn(S)-L catalysts, the adsorption free energy ($G_{ad}$) of the competing intermediates on the derived S-doped Sn catalysts surface was then calculated[51,52]. Herein, various possible adsorption configurations were considered, and the most stable adsorption structure for *OCHO was taken as the active site (Supplementary Fig. 26). For HCOOH production, the formation of *OCHO intermediates was considered as a determining step via proton-coupled electron transfer (PCET) process[4,19]. Compared with the pure Sn surface, the binding of the carbon-containing species *OCHO is significantly stronger on S-doped Sn, while *H adsorption is much weaker (Fig. 5d), indicating the suppression of HER after S doping. This data is also in accordance with the high *OCHO coverage on Sn(S)-H observed in the in situ ATR-FTIR (Fig. 5a, b). In addition, $G_{ad}$(*OCHO) is much lower than $G_{ad}$(*COOH), i.e., the intermediate to generate HCOOH is more energetically favorable than that to generate CO, which is consistent with the observed strong peak for *OCHO on Sn(S)-H catalyst in the in situ SERS and in situ ATR-FTIR spectra (Fig. 5a, b).

The free energy diagram shows the mechanism for $CO_2RR$ producing HCOOH (Fig. 5e). Under U = −1.1 V vs. SHE, the reaction is basically a readily exothermic process. The stronger adsorption of *OCHO on S-doped Sn well explains the high selectivity to HCOOH at the catalyst at high current density (Fig. 4a). The pathway via *OCHO is energetically more favorable than *COOH (Supplementary Fig. 27), which confirms that the reaction selectively produces HCOOH rather than CO. To better understand the influences of the doping S atoms on various adsorption species, a differential charge density analysis was carried out between Sn (100) surface with and without S-doping (Fig. 5f). When a subsurface Sn atom is replaced by S, the electron density on the surface is significantly reduced (blue area), indicating that the Sn bonded to S in the surface layer is positively charged. Therefore, such surface active sites are more inclined to interact with the electron-rich oxygen-containing groups, especially the *OCHO with two O atoms binding with the surface, rather than the proton *H. The theoretical calculations above imply that S-doped Sn promotes $CO_2RR$ to produce HCOOH but suppresses HER. Thus, the modestly increased S amount in Sn(S)-H demonstrates a better $CO_2RR$ performance than Sn(S)-L.

## Discussion

In summary, we report a phase engineering strategy of π-SnS that can stabilize rich S dopants on Sn subsurface in acidic medium for efficient $CO_2$-to-HCOOH production. The stabilized Sn-S active sites modulate $CO_2RR$ intermediates coverage, achieving acidic $CO_2$-to-HCOOH electrolysis (pH = 3) with industrial-level current densities. The π-SnS derived S-doped Sn catalyst achieves a high FE (over 70%) of HCOOH production with current densities from −0.1 to −1 A cm$^{-2}$ in acidic medium. Furthermore, the HCOOH production rate can reach the industrial level of 13.7 mmol h$^{-1}$ cm$^{-2}$, superior to most reported electrocatalysts. FT-EXAFS, in situ Raman spectroscopy and theoretical simulations reveal that π-SnS has a stronger intrinsic binding strength of Sn-S than conventional α-SnS with shorter Sn-S bond, which derives higher S content in the Sn subsurface (Sn(S)-H) after $CO_2RR$ activation under high current density. The Sn-S active sites have strong *OCHO adsorption and weakened *H adsorption, leading to a more energetically favorable $CO_2$-to-HCOOH pathway for Sn(S)-H, unveiled by in situ SERS, in situ ATR-FTIR and DFT calculations. More importantly, the Sn-S active site promotes *OCHO coverage at high current densities, leading to stable and high-rate production of HCOOH. Our work demonstrates the importance of phase-engineering in optimizing catalysts for high-performance acidic $CO_2RR$ and provides a promising route toward the efficient electrosynthesis of value-added chemicals at industrial current density.

## Methods

### Materials preparation

Synthesis of π-SnS nanocrystals is conducted by a typical hydro-thermal method. 0.22 g SnCl$_2$·2H$_2$O and 0.5 g polyvinylpyrrolidone (PVP, $M_w$ = 1,300,000) were dissolved in 40 mL of deionized (DI) water to form a clear solution under magnetic stirring. After that, 0.11 g cysteine (L-Cys) was added and stirred at room temperature for 40 min. The solution was transferred to a Teflon-lined stainless auto-clave and maintained at 200 °C for 110 min. After the reaction, the solution was cooled to room temperature, and the final product was collected through centrifugation and washed with DI water and etha-nol several times. The normal α-SnS was purchased from Sigma-Aldrich (Saint Louis, Missouri, USA).

### Electrochemical measurement

A three-electrode flow cell was used to investigate the $CO_2RR$ catalytic activity of different samples. For working electrode preparation, cata-lyst inks were made by mixing 4 mg catalysts with 720 μL isopropanol, 240 μL DI water and 40 μL of 5 wt% Nafion solution. After sonicating for 1 h, the inks were airbrushed onto a square (1 × 1 cm$^2$) hydrophobically treated carbon paper as the cathode electrode on a heating plate (70 °C). Ag/AgCl electrode (stored in saturated KCl) and commercial IrTa alloy electrode were used as reference electrode and counter electrode, respectively. Nafion 117 was used as proton exchange mem-brane to separate the working electrode and counter electrode.

$CO_2$ reduction performance was tested using a chron-opotentiometry method by applying different current densities. 0.5 M K$_2$SO$_4$ + H$_2$SO$_4$ (pH = 3) was chosen as the electrolyte. The pH value of the electrolyte was determined by a pH meter. The anode and cathode chamber volumes were both 20 mL. The $CO_2$ flow rate was 80 sccm controlled by a mass flow controller, and the electrolyte flow rate was stabilized at 5 mL min$^{-1}$. The duration of each chronopotentiometry test was 1000 s. Before test, a pre-reduction process was performed for each catalyst at −0.1 A cm$^{-2}$ for 900 s. The resistance of 2.4 Ω × 85% was used to calculate the iR-correction. The potential of the working electrode was converted to RHE reference scale by the following equation:

$$E_{RHE} = E_{Ag/AgCl} + 0.0591 \times pH + 0.197(V) \qquad (1)$$

Gas products were collected from 200 s to 800 s during each chronopotentiometry test and determined using gas chromatography (GC, Agilent 8890 GC System). Liquid products were detected using $^1$H nuclear magnetic resonance ($^1$H NMR, Bruker 500 M), and dimethyl sulfoxide (DMSO) was used as the internal standard.

The Faradaic efficiency of H$_2$, CO and HCOOH was calculated as follows:

$$FE = eF \times n/Q = eF \times n/(I \times t) \times 100\% \qquad (2)$$

Where e is the number of electrons transferred (for H$_2$, CO and HCOOH determined as 2), F is the Faraday constant, Q is the charge, I is current, t is the running time and n is the amount of product (in moles) determined by GC or $^1$H NMR.

SPCE for HCOOH was calculated at 25 °C, 1 atm according to the following equation:

$$SPCE_{HCOOH} = (60s \times n)/[flow\ rate(sccm) \times t(s) \div 24.05(l/mol)] \qquad (3)$$

Where n is the amount of HCOOH (in moles) determined from $^1$H NMR, and the running time (t) is 1000 s for each flow rate.

### In situ ATR-FTIR measurement

In situ ATR-FTIR was performed on Thermo-Fisher Nicolet iS20 with liquid N$_2$ cooled HgCdTe (MCT) detector and a VeeMax III (PIKE

technologies) accessory. A custom-made three-electrode electrochemical single-cell was selected for electrochemical tests. For working electrode, catalysts were sprayed on the Ge prism with a fixed angle (60°) and polished reflecting plane (0.05 μm, Kemet. Int. Ltd.). A Pt-wire and a saturated Ag/AgCl electrode were used as counter and reference electrodes, respectively. 0.5 M $K_2SO_4$ + $H_2SO_4$ (pH = 3) was used as electrolyte with constantly purged $CO_2$ (flow rate: 10 sccm). For ATR-FTIR measurement, 32 scans were collected with a spectral resolution of 4 cm$^{-1}$ for each spectrum, and open circuit potential (OCP) was recorded as a comparison, chronoamperometric tests from 0 to −1.9 V (vs. RHE) to collect data.

## In situ Raman measurement

In situ Raman measurements were divided for the detection of reaction intermediates and derived catalysts. For In situ SERS toward reaction intermediates were conducted on a HORIBA LabRAM HR Evolution Raman spectrometer with a 633 nm solid laser as an excitation source. A homemade flow cell with a quartz window was used to collect the Raman signal from GDE (sprayed with catalyst). The electrolyte was 0.5 M $K_2SO_4$ + $H_2SO_4$ (pH = 3), and $CO_2$ was introduced to the back of GDE in 80 sccm. Ag/AgCl electrode and commercial IrTa alloy electrode were used as reference electrode and counter electrode, respectively. For In situ Raman spectra of derived catalysts, HORIBA LabRAM HR Evolution Raman spectrometer with a 60X (1.0 N.A) water-immersion objective (Olympus) was performed, and the laser wavelength was 532 nm. A screen-printed electrode (Pine, RRPE1002C) was used as the electrode with an applied potential of −1.5 V (vs.RHE). $CO_2$-saturated 0.5 M $K_2SO_4$ solution (pH = 3) was used as electrolyte.

## Material characterization

XRD data were collected on a Rigaku MiniFlex 600 X-ray diffractometer. SEM images were obtained on a FEI QUANTA 450 electron microscope. The TEM images, aberration-corrected TEM images, HAADF-STEM imaging and EDS mapping were taken on a FEI Titan Themis 80-200 operating at 200 kV. X-ray photoelectron spectroscopy (XPS) analysis was carried out under ultrahigh vacuum (degree of vacuum about 5*10$^{-9}$ m bar) on Thermo Scientific K-Alpha+ with mono Al Kα source in Voltage of 15 kV and energy of 1486.6 eV.

Synchrotron-based NEXAFS measurements were determined on the soft X-ray spectroscopy beamline at the Australian Synchrotron, which is equipped with a hemispherical electron analyzer and a microchannel plate detector to permit concurrent recording of the total electron, and partial electron yield. Calibration of XPS data was normalized to the photoelectron current of the photon beam, measured on an Au-grid. Raw XANES data were normalized using Igor Pro 8 software.

## DFT calculations

DFT calculations were performed using Vienna ab initio simulation packages (VASP)[53]. All calculations used the projector-augmented wave (PAW) pseudopotentials and the revised Perdew-Burke-Ernzerhof (RPBE) exchange-correlation functional, with the DFT-D3 method for van der Waals corrections[54–57]. The plane wave cutoff was set to 600 eV. All atoms were fully relaxed until the convergence of energy and forces were $1 \times 10^{-5}$ eV and 0.03 eV·Å$^{-1}$, respectively. A $4 \times 4 \times 1$ and a $6 \times 6 \times 1$ grid of k-points were used to sample the first Brillouin zones of the surfaces for structural optimizations and charge density analysis, respectively. The VASPsol package was used for implicit solvation corrections[58].

## Ab initio thermodynamics approach

The crystal structure of π-SnS (Supplementary Fig. 28) was built according to previous reported π phase of SnS structure. The structural symmetry is slightly improved for ease of theoretical modelling.

The α-SnS was used for comparison. Corresponding to the particles with uniform dodecahedron morphology observed in HAADF-STEM experiments (Fig. 2d–f), the (110) facet was used for calculation. The Pourbaix diagrams were calculated by evaluating the surface free energy γ. For stoichiometric surface or with Sn- and S-terminations, the composition is considered as $Sn_xS_y$. Within the ab initio thermodynamics approach, γ is defined as:

$$\gamma = \frac{1}{2A}[G_{surf}(Sn_xS_y) - x\mu_{bulk}(SnS) - (y-x)\mu_S] \quad (4)$$

$G_{surf}(Sn_xS_y)$ is the Gibbs free energy of the surface candidate structure, described via a symmetric slab model with two equivalent surfaces of surface unit-cell area $A$ and containing x Sn atoms and y S atoms. $\mu_{bulk}(SnS)$ is the DFT-calculated energy of bulk π-SnS or α-SnS. $\mu_S$ is the chemical potential of S species, which is reference to the $H_2S$ and $H_2$ molecules:

$$\mu_S = G_{H_2S(molecule)} - 2\mu_H \quad (5)$$

$$\mu_H = \frac{1}{2}G_{H_2(molecule)} + eU_{SHE} - k_BT\log_{10}(pH) \quad (6)$$

$\mu_H$ depends on the bias potential (U) and pH conditions.

The Gibbs free energy of molecules was DFT-calculated energy with vibrational corrections:

$$G_{(molecule)} = E_{(molecule)} + F_{molecule}^{vib} \quad (7)$$

## Adsorption free energy calculations

A symmetric slab model of the $(2 \times 2)$ Sn (100) surface was used to calculate the adsorption free energies ($G_{ad}$) of the possible intermediates *H, *OCHO, *COOH and *HCOOH. One Sn atom at the subsurface was replaced with S atom to simulate the S-doped surface structure (Supplementary Fig. 29). $G_{ad}$ for different adsorption species was calculated by:

$$G_{ad} = E_{ad} + ZPE + \int C_p dT - TS \quad (8)$$

where $E_{ad}$ is the adsorption energy, ZPE is the zero-point energy, $\int C_p dT$ is the enthalpic temperature correction and $TS$ is the entropic correction. $E_{ad}$ is calculated as:

$$E_{ad} = E_{total} - E_{surf} - \mu_{adsorbate} \quad (9)$$

where $E_{total}$ is the DFT total energy of the surface with the adsorbate, $E_{surf}$ is the energy of the corresponding pristine surface and $\mu_{adsorbate}$ is the chemical potential of the adsorbate species referenced to $H_2(g)$ and $CO_2(g)$. As defined, more negative $G_{ad}$ values indicate a stronger binding. In the free energy diagram, the free energies under a certain potential (G(U)) were approximately accounted for within the thermodynamic computational hydrogen electrode (CHE) approach for the PCET steps, i.e.:

$$CO_2 \rightarrow {}^*OCHO \text{ and} {}^*OCHO \rightarrow {}^*HCOOH : G(U) = G_{ad} + neU \quad (10)$$

where e is the elementary negative charge, n is the number of charge transferred[59].

## Data availability

Data that support findings from this study are available from the corresponding author on reasonable request.

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

## Acknowledgements

This work was financially supported by the Australian Research Council through Discovery Project Programs (FL170100154, DP220102596). J.D. acknowledges financial support from the Basic Science Center Program for Ordered Energy Conversion of the National Natural Science Foundation of China (NSFC 51888103 and 52006105). H.J. gratefully acknowledges financial support from Institute for Sustainability, Energy and Resources, The University of Adelaide, Future Making Fellowship. H.S. acknowledges S. Zhang, H. Li, P. Tang and M. Zheng from School of Chemical Engineering, A. Slattery and S. Gilbert from Adelaide Microscopy at the University of Adelaide for their help. NEXFAS measurements were undertaken on the soft X-ray beamline at the Australian Synchrotron. XAS spectra of Sn K-edge were acquired on HXMA beamline at Canadian Light Source Inc, Canada. SEM and TEM measurements were conducted at Adelaide Microscopy, The Centre for Advanced Microscopy and Microanalysis.

## Author contributions

S.-Z.Q., H.J., and H.S. conceived the project. S.-Z.Q. supervised the project and whole studies. H.S. and H.J. designed and carried out experiments. H.S. conducted materials synthesis. H.S. and H.W. conducted electrochemical measurements. H.L. performed the DFT computations. H.S. and H.J. conducted in situ ATR-FTIR measurements and TEM characterizations. Y.J. and J.D. assisted in the analysis of findings. H.S., H.J., and S-Z.Q. wrote the paper. S-Z.Q. reviewed and edited the paper. H.S., H.J., and H.L. contributed equally to this work. All authors have approved the final version of the paper.

## Competing interests

The authors declare no competing interests.
