## [Peer review file · Nature Communications]

REVIEWER COMMENTS

Reviewer #1 (Remarks to the Author):

The manuscript by Shen et al. demonstrates a phase engineering strategy to steer the construction of Sn active sites toward acidic CO₂-to-HCOOH process. In situ characterizations and theoretical calculations evidence the stronger Sn-S binding strength on π -SnS phase than that on the conventional α -SnS phase, facilitating the stabilization of residual S species. The S dopants finally contribute to the efficient and highly selective CO₂ conversion to HCOOH over derived Sn(S)-H catalyst. The presented results are interesting given that the importance of acidic CO₂RR subfield. Mechanically, the identification or manipulation of active sites is a central topic in catalysis science, and the CO₂RR catalyst stability deserves an in-depth concern especially in acidic medium. However, there are some inadequacies required to be first clarified; therefore I would like to recommend a major revision before acceptance to address the concerns listed below:

1. From my perspective, the sulfate ions in acidic electrolyte could disturb the precise analysis of S element in Sn(S)-H catalyst after activation step (NEXAFS and XPS in Fig. 2e, f), although it is claimed by authors that sulfate ions and Sn-S can be distinguished. In this case, an acidic electrolyte without S element could be better for reaching a more convincing conclusion. Besides, according to XRD in Fig. 2a, SnS phase almost disappears in metallic Sn(S)-H catalyst, while for Raman spectra in Fig. 2d, some SnS phase still exists even after 10 h at -1.5 V (vs RHE). How to explain this seeming contradiction?
2. In Fig. 3d, the stability test lasts only for 260 min, which hardly meets the industrial demands. Therefore, the authors should attempt to prolong the test, and also provide the TEM image of Sn(S)-H after durability test to inspect its structural integrity. In addition, is S element stable in Sn(S)-H catalyst after activation step and during electrochemical CO₂RR process? Experimentally, what is the role of acidic environment on influencing the reconstruction of π -SnS or Sn(S)-H catalyst during activation or CO₂RR working process?
3. The authors pay attention on inhibiting the sulfur dissolution, which is also considered by another latest work (ACS Catal. 2022, 12, 13533-13541). However, for acidic CO₂RR, I would stress that metal dissolution should be more important in terms of catalyst stability in acidic medium. Given that the redox potential of Sn element, I recommend performing ICP test of the electrolyte to examine the possible Sn dissolution after acidic CO₂RR operation. On the other hand, does the remained S element affect the stability of Sn(S)-H or Sn(S)-L catalyst? This might be answered by comparing the stability performance of Sn(S)-H or Sn(S)-L. Furthermore, the DFT calculations can give some proofs by evaluating the parameters such as vacancy formation energy difference of Sn atom while after introducing S element.
4. I suggest that authors should discreetly check the Raman data in Supplementary Fig. 19, especially for the analysis on symmetric/asymmetric stretching vibration (vsO-C-O) of *OCHO intermediate at 1350 and 1580 cm⁻¹, claimed by them. These two bands are more likely attributed to the D and G band of graphene or carbon materials (Nature Nanotechnol., 2013, 8(4): 235-246), as a carbon-based GDE is used during in-situ electrochemical measurements.

5. I understand that the authors attribute the catalytic performance difference to the S element for Sn(S)-H or Sn(S)-L. Despite this, they should try to exclude other factors, such as defects, or coordination environment of active sites. In atomic-resolution HAADF-STEM (Fig. 2b and Supplementary Fig. 7), the exposed facet with corresponding lattice parameters should be labeled to make a clear comparison. By the way, the whole manuscript should be carefully checked to correct those possible expression errors. Finally, I suggest incorporating more related literatures to enrich the background or discussion on structural reconstruction or catalyst stability topic, such as Adv. Funct. Mater. 2022, 32, 2111193; Nano Res., 2022, 15(4): 3283-3289; Adv. Energy Mater. 2022, 12, 2200970, etc.

Reviewer #2 (Remarks to the Author):

The authors report a phase engineering strategy of π -SnS that can stabilize rich S dopants on Sn subsurface in acidic medium for efficient CO₂-to-HCOOH production.

The π -SnS derived S-doped Sn catalyst achieves a high FE (over 70 %) of HCOOH production. The topic is interesting and the results are reliable. It might be accepted after the following issues are addressed.

1. It is stated that "As shown in Fig. 4a, the distinct *OCHO signal around 1367 cm⁻¹ on Sn(S)-H confirms the efficient HCOOH generation." Actually, the *OCHO hydrogenation is possible to generate other intermediate.
2. I recommend to check the adsorption energy of *COOH on the surface, and the reaction path *CO₂→*COOH→.....should be shown in the Free energy diagram of CO₂ reduction.
3. According to the *OCHO with two O atoms binding with the surface, rather than the proton *H, the authors conclude that S-doped Sn promotes CO₂RR to produce HCOOH but suppresses HER. Did the authors try all possible adsorption sites for *OCHO and *H? please present the adsorption energies and ΔG for *OCHO and *H adsorption.

Response to Reviewer #1

Reviewer's Remarks to Authors

The manuscript by Shen et al. demonstrates a phase engineering strategy to steer the construction of Sn active sites toward acidic CO₂-to-HCOOH process. In situ characterizations and theoretical calculations evidence the stronger Sn-S binding strength on π -SnS phase than that on the conventional α -SnS phase, facilitating the stabilization of residual S species. The S dopants finally contribute to the efficient and highly selective CO₂ conversion to HCOOH over derived Sn(S)-H catalyst. The presented results are interesting given that the importance of acidic CO₂RR subfield. Mechanically, the identification or manipulation of active sites is a central topic in catalysis science, and the CO₂RR catalyst stability deserves an in-depth concern especially in acidic medium. However, there are some inadequacies required to be first clarified; therefore I would like to recommend a major revision before acceptance to address the concerns listed below.

Response

We thank Reviewer #1 for his/her valuable comments and positive recommendation.

Comment 1-1

From my perspective, the sulfate ions in acidic electrolyte could disturb the precise analysis of S element in Sn(S)-H catalyst after activation step (NEXAFS and XPS in Fig. 2e, f), although it is claimed by authors that sulfate ions and Sn-S can be distinguished. In this case, an acidic electrolyte without S element could be better for reaching a more convincing conclusion. Besides, according to XRD in Fig. 2a, SnS phase almost disappears in metallic Sn(S)-H catalyst, while for Raman spectra in Fig. 2d, some SnS phase still exists even after 10 h at -1.5 V (vs RHE). How to explain this seeming contradiction?

Response

We agree with Reviewer #1 that the S species in potassium sulfate and sulfuric acid may affect the spectroscopic data. Therefore, we conducted a control experiment using potassium perchlorate and perchloric acid as alternatives. We didn't use phosphoric acid and nitric acid because these two acids reveal either incomplete ionization or NO₃⁻ reduction reaction on the cathode. However, due to the low solubility of potassium perchlorate, we used saturated potassium perchlorate (around 0.12 M) mixed with perchloric acid (pH = 3) solution and 1 M sodium perchlorate mixed with perchloric acid (pH = 3) solution, respectively.

As shown in **Figure R1**, π -SnS derived catalysts (Sn(S)-H) illustrate higher residual S amount than α -SnS derived catalysts (Sn(S)-L) in both saturated $\text{KClO}_4 + \text{HClO}_4$ electrolyte and 1 M $\text{NaClO}_4 + \text{HClO}_4$ electrolyte, confirming the stronger Sn-S binding strength on π -SnS than α -SnS. These results are in accordance with that in 0.5 M $\text{K}_2\text{SO}_4 + \text{H}_2\text{SO}_4$ electrolyte. However, we still found sulfates (SO_4^{2-}) and sulfites (SO_3^{2-}) in Sn(S)-H and Sn(S)-L. Because no SO_4^{2-} or SO_3^{2-} is detected in π -SnS and α -SnS pre-catalyst powders (**Figure R2** and **Supplementary Fig. 8**), we presume that the SO_4^{2-} and SO_3^{2-} (**Figure R1**) originate from two pathways: 1) 5 wt % Nafion solution (40 μL / mL ink), as the high valence state of S (+6) existed in perfluorinated resin; 2) the oxidation of dissociated S (from pre-catalysts) after CO_2RR activation.

Figure R1. High-resolution S 2p XPS spectra of Sn(S)-H and Sn(S)-L after CO_2RR activation in (a) saturated $\text{KClO}_4 + \text{HClO}_4$ electrolyte and (b) 1M $\text{NaClO}_4 + \text{HClO}_4$ electrolyte.

Figure R2. High-resolution S 2p XPS spectra of (a) π -SnS and (b) α -SnS pre-catalysts.

In response to address this comment, we have in our:

1) R-SI, added **Figure R1** as **Supplementary Fig. 9** with the following clarifying note;

*‘Note: We use the electrolyte without sulfur element to eliminate the influence of electrolyte on Sn-S detection. Potassium perchlorate (KClO₄) + perchloric acid (HClO₄) is a suitable choice. However, due to the low solubility of KClO₄, we use saturated potassium perchlorate (around 0.12 M) + perchloric acid (pH = 3) solution and 1 M sodium perchlorate + perchloric acid (pH = 3) solution for comparison. π -SnS derived catalysts (Sn(S)-H) illustrate higher residual S amount than α -SnS derived catalysts (Sn(S)-L) in both saturated KClO₄ + HClO₄ electrolyte and 1 M NaClO₄ + HClO₄ electrolyte, confirming the stronger Sn-S binding strength on π -SnS phase than α -SnS. The results are in accordance with that in 0.5 M K₂SO₄ + H₂SO₄ electrolyte. However, we can still find residual sulfates (SO₄²⁻) and sulfites (SO₃²⁻) in Sn(S)-H and Sn(S)-L. The SO₄²⁻ and SO₃²⁻ detected in **Supplementary Fig. 9** could originate from two pathways: 1) Nafion binder with high valence state of S (+6); 2) the oxidation of dissociated S (from pre-catalysts) after CO₂RR activation.’*

2) R-MS, p. 10, para. 1, included the additional text as follows;

*‘The similar evidence of higher residual S amount in Sn(S)-H could also be found in other sulfur-free acidic electrolytes (**Supplementary Fig. 9**).’*

The XRD mainly detects the crystal structure of the material. However, the subsurface residual S-Sn bonds in Sn(S)-H after activation can't be detected by XRD because the reconstruction induced lattice collapse. Therefore, we use *in situ* Raman spectroscopy to detect the vibration of residual S-Sn bond (Nat. Protoc. 2023, <https://doi.org/10.1038/s41596-022-00782-8>). Compared with XRD, *in situ* Raman spectroscopy has a much lower detection depth but is more surface sensitive even though there is no long-range ordered SnS lattice. Therefore, after activation under -0.1 A cm⁻² for 900 s, XRD reveals a metallic Sn structure in Sn(S)-H, while the residual S species in the Sn subsurface (Sn-S) are detected by *in situ* Raman spectra.

Comment 1-2

In Fig. 3d, the stability test lasts only for 260 min, which hardly meets the industrial demands. Therefore, the authors should attempt to prolong the test, and also provide the TEM image of Sn(S)-H after durability test to inspect its structural integrity. In addition, is S element stable in Sn(S)-H catalyst after activation step and during electrochemical CO₂RR process? Experimentally, what is the role of acidic environment on influencing the reconstruction of π -SnS or Sn(S)-H catalyst during activation or CO₂RR working process?

Response

We found that the stability of flow cells is affected by the potassium sulfate deposition on the gas-diffusion layer (Nat. Catal., 2022, 5, 564-570.). Therefore, we conducted the test by periodically removing the potassium sulfate deposition and refreshed the electrolyte for both Sn(S)-H and Sn(S)-L. In addition, the volume of the cathode and anode electrolytes are changed to 2.2 L and 1 L, respectively, to ensure a stable electrocatalytic environment. As seen in **Figure R3**, Sn(S)-H maintained a high FE of HCOOH (over 85%) after 13.5 h under a current density of -400 mA cm^{-2} . Further HAADF-STEM images of Sn(S)-H confirm the stable Sn structure after stability measurement (**Figure R4**).

In addition, both EDS mapping and XPS spectra of Sn(S)-H after stability measurement confirm the stable S-Sn bonding on Sn(S)-H, shown in **Figure R4b** and **Figure R5a**. For comparison, the Sn(S)-L shows a trace amount of S-Sn bond after stability measurement, which is much lower than that in Sn(S)-H, **Figure R5b**.

In response to address this comment, we have in our:

1) R-MS, put **Figure R3** as **Fig. 3d**, changed the corresponding Figure caption as '*Stability measurement of HCOOH production at the total current density of -400 mA cm^{-2} for Sn(S)-H.*'

2) R-SI, added **Figure R4** and **Figure R5** as **Supplementary Figs. 21** and **22**, respectively.

3) R-MS, p. 13, para. 1, added the text as follows;

'The Sn(S)-H shows a 85 % FE of HCOOH production under -400 mA cm^{-2} in 13.5 h stability measurement (Fig. 3d). In addition, Sn(S)-H maintained metallic Sn structure with stable S-Sn bonding after stability measurement (Supplementary Figs. 21, 22).''

Figure R3. Stability measurement of HCOOH production at the total current density of -400 mA cm^{-2} for Sn(S)-H.

Figure R4. Structural characterization of Sn(S)-H after stability measurement. **a**, TEM image. **b**, EDS mapping images of Sn and S elements. **c**, HAADF-STEM image. **d**, Magnified HAADF-STEM image taken from the corresponding area in (c).

Figure R5. High-resolution S 2p XPS spectra of (a) Sn(S)-H and (b) Sn(S)-L after stability measurement with the corresponding pre-catalysts π -SnS (a) and α -SnS (b) inserted in top.

To understand the role of acidic environment in the reconstruction of π -SnS, we used neutral (0.5 M K_2SO_4) and alkaline electrolyte (1M KOH) to detect the catalysts evolution. As demonstrated in **Figure R6**, the XPS spectrum of the sample derived in neutral electrolyte reveals a higher residual S (Sn-S) amount than that in acidic and alkaline environments. The lower residual S species in Sn(S)-H derived from acidic or alkaline environment could relate to the fast hydrogen evolution reaction (HER) dynamics, which accelerates the S dissolution during the reconstitution. However, acidic media is ideal for HCOOH production and was used as the electrolyte in this work.

Figure R6. High-resolution S 2p XPS spectra of Sn(S)-H after CO₂RR activation in (a) 0.5 M K_2SO_4 and (b) 1M KOH electrolytes.

Comment 1-3

The authors pay attention on inhibiting the sulfur dissolution, which is also considered by another latest work (ACS Catal. 2022, 12, 13533-13541). However, for acidic CO₂RR, I would stress that metal dissolution should be more important in terms of catalyst stability in acidic medium. Given that the redox potential of Sn element, I recommend performing ICP test of the electrolyte to examine the possible Sn dissolution after acidic CO₂RR operation. On the other hand, does the remained S element affect the stability of Sn(S)-H or Sn(S)-L catalyst? This might be answered by comparing the stability performance of Sn(S)-H or Sn(S)-L. Furthermore, the DFT calculations can give some proofs by evaluating the parameters such as vacancy formation energy difference of Sn atom while after introducing S element.

Response

Thanks for the valuable suggestion. We took the ICP test for the electrolyte at different reaction time for Sn(S)-H and Sn(S)-L (**Figure R7**). In contrast to Sn(S)-L, Sn(S)-H demonstrated a lower dissolving rate, proving that the strong S-Sn bond can prevent the dissolution of Sn in Sn(S)-H.

Figure R8 shows the stability performance of Sn(S)-H and Sn(S)-L, the FE of HCOOH in Sn(S)-L decreased to 40% after 6 h, while Sn(S)-H maintained a high FE of HCOOH (over 85%) after 13.5 h, which could be attribute to the remained S element. Therefore, the lower Sn dissolving rate in Sn(S)-H confirms the positive effect of S for acid resistance.

In response to fully address this comment of Reviewer #1 we have in our:

1) R-SI, added **Fig. R7** and **Fig. R8b** as **Supplementary Figs. 23**

2) R-MS, p. 13, para. 1, added the text as follows;

*‘For acidic CO₂ reduction, metal dissolution with related long-term stability are important aspects that cannot be ignored. The Sn(S)-H shows a 85 % FE of HCOOH production under -400 mA cm^{-2} in 13.5 h stability measurement (**Fig. 3d**). In addition, Sn(S)-H maintained metallic Sn structure with stable S-Sn bonding after stability measurement (**Supplementary Figs. 21, 22**). While Sn(S)-L exhibited continuously decreasing FE of HCOOH for 6 h with distinctly higher Sn dissolving ratio than Sn(S)-H (**Supplementary Fig. 23**), which could be attributed to the desolution of S-Sn sites.’*

3) In R-MS, cited this paper in References;

*‘RF 39. Liu, F. et al. Inhibiting sulfur dissolution and enhancing activity of SnS for CO₂ electroreduction via electronic state modulation. *ACS Catal.* **12**, 13533-13541 (2022).’*

Figure R7. Dissolving ratio of Sn(S)-H and Sn(S)-L at different reaction time during the stability measurement.

Figure R8. Stability measurement of HCOOH production at the total current density of -400 mA cm^{-2} for Sn(S)-H (a) and Sn(S)-L (b).

Comment 1-4

I suggest that authors should discreetly check the Raman data in Supplementary Fig. 19, especially for the analysis on symmetric/asymmetric stretching vibration ($\nu_{\text{S}}\text{O-C-O}$) of $^*\text{OCHO}$ intermediate at 1350 and 1580 cm^{-1} , claimed by them. These two bands are more likely attributed to the D and G band of graphene or carbon materials (Nature Nanotechnol., 2013, 8(4): 235-246), as a carbon-based GDE is used during *in-situ* electrochemical measurements.

Response

Thanks for the valuable suggestion. We double-check the *in situ* SERS of $^*\text{OCHO}$ intermediate in **Supplementary Fig. 19**. Although some related studies have identified the potential-dependent broad peaks at 1350 and 1590 cm^{-1} as the symmetric stretching vibration ($\nu_{\text{S}}\text{O-C-O}$) and the asymmetric stretching vibration ($\nu_{\text{AS}}\text{O-C-O}$) in $^*\text{OCHO}$ (ACS Catalysis, 2022, 12, 8601-8609; ACS Catalysis, 2020, 10, 8601-8609), carbon-based materials also exhibit prominent peaks at the same positions (Nature Nanotechnology, 2013, 8(4), 235-246). Therefore, we deleted the labels of the broad peaks around 1350 and 1590 cm^{-1} in **Figure R9**. In addition, the $^*\text{OCHO}$ intermediate is widely believed as the main intermediate of formate/formic acid production for Sn and Sn-based materials (Joule, 2017, 1, 794-804; Nat. Comm., 2022, 13, 2486; Angew. Chem. Int. Ed., 2021, 60, 26233-26237), which is further confirmed in our *in situ* ATR-FTIR spectra (**Fig. 4a, b** and **Supplementary Fig. 25**). Therefore, we believe that deleting these two labels does not affect the accuracy of the conclusion.

In response to address directly this comment, we have in our:

1) In R-SI, revised **Supplementary Fig. 24** to delete the marks of the broad peaks around 1350 and 1590 cm^{-1} .

2) In R-MS, p.14, para. 2, the description of Raman data changed to the following text;
'The narrow peak at 1550 cm^{-1} corresponds to asymmetric stretching vibration ($\nu_{\text{as}}\text{O-C-O}$) of $^\text{OCHO}$. Since no obvious $^*\text{COOH}$ adsorption peaks were detected, $^*\text{OCHO}$ was determined as the primary intermediate for acidic CO_2 -to- HCOOH in our work, which agrees with the main intermediate of HCOO^- formation in neutral and alkaline media.'*

3) In R-MS, cited the following papers in References;

'RF 48. Ferrari A. C. & Basko D. M. Raman spectroscopy as a versatile tool for studying the properties of graphene. Nat. Nanotechnol. 8, 235-246 (2013).'

'RF 49. Ren, B. et al. Nano-crumpled induced Sn-Bi bimetallic interface pattern with moderate electron bank for highly efficient CO_2 electroreduction. Nat. Commun. 13, 2486 (2022).'

Figure R9. In situ SERS of Sn(S)-H under CO_2 RR with increased potential in flow cell. **a**, Low potential (OCP to -0.6 V). **b**, High potential (-1.4 to -2 V).

Comment 1-5

I understand that the authors attribute the catalytic performance difference to the S element for Sn(S)-H or Sn(S)-L. Despite this, they should try to exclude other factors, such as defects, or coordination environment of active sites. In atomic-resolution HAADF-STEM (Fig. 2b and Supplementary Fig. 7), the exposed facet with corresponding lattice parameters should be labeled to make a clear comparison. By the way, the whole manuscript should be carefully checked to correct those possible expression errors. Finally, I suggest incorporating more related literatures to enrich the background or discussion on structural reconstruction or catalyst stability topic, such as Adv. Funct. Mater. 2022, 32, 2111193; Nano Res., 2022, 15(4): 3283-3289; Adv. Energy Mater. 2022, 12, 2200970, etc.

Response

We conducted additional HAADF-STEM measurements to explore the influence of defects, vacancies and coordination environment on the electrocatalysts (Adv. Funct. Mater., 2022, 32, 2111193; Angew. Chem. Int. Ed., 2021, 60, 18178-18184; Nat. Comm., 2021, 12, 660). We found that no obvious defects were observed in both Sn(S)-H and Sn(S)-L (**Figure R10**). Thus, we believe the main contribution of the catalytic performance difference for Sn(S)-H and Sn(S)-L is the different residual S amount.

In response to address this, we have, in our:

- 1) R-SI, added **Figure R10** as **Supplementary Fig. 10**.
- 2) R-MS, p.10, para. 1, included the following clarifying text;

*'Due to the complexity of the reconstitution to the derived catalysts, we also checked other potential factors such as defects which could affect the catalytic performance. As no obvious defects were observed in both Sn(S)-H and Sn(S)-L (**Supplementary Fig. 10**), we confirm the main difference after reconstitution between Sn(S)-H and Sn(S)-L is the amount of residual S.'*

Figure R10. HAADF-STEM images of Sn(S)-H (a,b) and Sn(S)-L (c,d).

To make a clear comparison of the exposed facet for Sn(S)-H and Sn(S)-L, we use the magnified HAADF-STEM images of Sn(S)-H and Sn(S)-L as shown in **Figure R11** and **Figure R12**. The exposed facets of Sn(S)-H and Sn(S)-L in **Figure R11** and **Figure R12** were determined as Sn (200), which is consistent with the XRD results. However, according to HAADF-STEM images, both Sn(S)-H and Sn(S)-L illustrated polycrystalline structures after reconstruction. We find Sn (220) and Sn (101) facets for Sn(S)-H and Sn(S)-L. For details, please refer to **Figure R13**.

Figure R11. Magnified HAADF-STEM image of catalyst derived from π -SnS with the inset of FFT pattern.

Figure R12. **a**, HAADF-STEM image of Sn(S)-L with the inset of FFT pattern. **b**, Magnified HAADF-STEM image taken from the corresponding area in (a).

Figure R13. **a**, HAADF-STEM image of Sn(S)-H with the magnified image (**b**). **c**, MHAADF-STEM image of Sn(S)-L with the magnified image (**d**).

We agree with Reviewer #1 to enrich the content of this article in background and discussion on structural reconstruction or catalyst stability, as structural reconstruction and catalyst stability are challenging areas with constant attention in current research of CO₂ reduction.

In response to address this comment directly, we have in our R-MS:

1) Cited the following papers in References;

‘RF 31. Y. et al. *In situ construction of thiol-silver interface for selectively electrocatalytic CO₂ reduction*. *Nano Res.* **15**, 3283-3289 (2021).’

‘RF 32. Yuan, Y. et al. *In situ structural reconstruction to generate the active sites for CO₂ electroreduction on bismuth ultrathin nanosheets*. *Adv. Energy Mater.* **12**, 2200970 (2022).’

‘RF 33. Lai, W., Ma, Z., Zhang, J., Yuan, Y., Qiao, Y. & Huang, H. *Dynamic evolution of active sites in electrocatalytic CO₂ reduction reaction: Fundamental understanding and recent progress*. *Adv. Funct. Mater.* **32**, 2111193 (2022).’

2) In R-MS, p. 8, para. 3, added the text to read;

'Derived catalysts typically display distinct structures from their pre-catalysts, however, this reconstitution process is often ignored. The identification of derived catalysts via reconstitution can contribute to a better understanding of the origin of catalytic performance.'

3) In R-MS, p. 13, para. 1, added the text as follows;

'For acidic CO₂ reduction, metal dissolution with related long-term stability are important aspects that cannot be ignored.'

Response to Reviewer #2

Reviewer's Remarks to Authors

The authors report a phase engineering strategy of π -SnS that can stabilize rich S dopants on Sn subsurface in acidic medium for efficient CO₂-to-HCOOH production.

The π -SnS derived S-doped Sn catalyst achieves a high FE (over 70 %) of HCOOH production. The topic is interesting and the results are reliable. It might be accepted after the following issues are addressed.

Response

We thank Reviewer #2 for his/her valuable comments and positive recommendation for publication.

Comment 2-1

*It is stated that "As shown in Fig. 4a, the distinct *OCHO signal around 1367 cm⁻¹ on Sn(S)-H confirms the efficient HCOOH generation." Actually, the *OCHO hydrogenation is possible to generate other intermediate.*

Response

We checked other possible intermediates after *OCHO hydrogenation. For the two *O atoms binding on the surface, further hydrogenation of *OCHO does not generate multi-carbon products as C-C coupling normally needs *C for binding atoms. Therefore, HCOOH is the thermodynamically optimal product, and H₂COOH* and H₂CO* are two possible intermediates to generate HCHO, CH₃OH or CO (Nat. Chem., 2014, 6, 320; Angew. Chem. Int. Ed., 2018, 57, 15045-15050; Nat. Nanotechnol., 2021, 16, 1386-1393.). According to the product detection and *in situ* experiments, no obvious H₂COOH* and H₂CO* intermediates are detected. Thus, we confirmed that HCOOH is the main product for *OCHO hydrogenation. To improve the rigor of the article, we have:

In R-MS, p. 14, para. 3, revised the text as follows;

*'For two CO₂-to-HCOOH intermediates, *OCHO is widely considered more efficient than *COOH for HCOOH production. As shown in Fig. 4a, the distinct *OCHO signal around 1367 cm⁻¹ on Sn(S)-H together with no obvious *OCHO hydrogenation intermediates detected in in situ ATR-FTIR experiments, confirm the efficient HCOOH generation of Sn(S)-H.'*

Comment 2-2

I recommend to check the adsorption energy of $*\text{COOH}$ on the surface, and the reaction path $*\text{CO}_2 \rightarrow *\text{COOH} \rightarrow \dots$ should be shown in the Free energy diagram of CO_2 reduction.

Response

The adsorption energy of $*\text{COOH}$ on the surface has been checked. The comparison in adsorption free energy of $*\text{COOH}$ and $*\text{OCHO}$ is shown in **Figure 4d**. In addition, we agree that a direct comparison of the reaction pathway *via* $*\text{OCHO}$ and $*\text{COOH}$ shows the reaction selectivity mechanism more intuitively. As shown in **Figure R14**, these two reaction pathways are plotted together. It can be seen that the $*\text{OCHO}$ pathway is energetically more favorable than $*\text{COOH}$. This well confirms that the reaction selectively produces HCOOH rather than CO .

In response to address this, we have, in our:

1) R-SI, added **Figure R14** as **Supplementary Fig. 27**.

2) R-MS, p.17, para. 2, included the following clarifying text:

'The pathway via OCHO is energetically more favorable than $*\text{COOH}$ (Supplementary Fig. 27), which confirms that the reaction selectively produces HCOOH rather than CO .'*

Figure R14. Free energy diagram of CO_2 electroreduction to HCOOH (solid steps, $\text{CO}_2 \rightarrow *\text{OCHO} \rightarrow *\text{COOH} \rightarrow \text{HCOOH}$), compared with to CO (dashed steps, $\text{CO}_2 \rightarrow *\text{COOH} \rightarrow *\text{CO} \rightarrow \text{CO}$), under $U = -1.1$ V (vs. SHE) on Sn (100) and S-doped Sn (100). The HCOOH pathway (through $*\text{OCHO}$) is significantly more favorable than the CO pathway (through $*\text{COOH}$).

Comment 2-3

According to the $*\text{OCHO}$ with two O atoms binding with the surface, rather than the proton $*\text{H}$, the authors conclude that S-doped Sn promotes CO_2RR to produce HCOOH but suppresses HER.

Did the authors try all possible adsorption sites for *OCHO and *H? please present the adsorption energies and ΔG for *OCHO and *H adsorption.

Response

Figure R15. Adsorption configuration and free energy (G_{ad}) of *OCHO and *H on Sn (100) and S-doped Sn (100). The adsorption site and G_{ad} shown in **Fig. 4d** is marked in bold. Steel blue sphere: Sn; yellow sphere: S; grey sphere: C; red sphere: O; white sphere: H.

We agree with Reviewer #2 that different possible adsorption sites need to be considered when calculating the adsorption energy. We have considered this point when comparing the adsorption free energies of *OCHO and *H in **Fig. 4d**. As shown in **Figure R15**, we calculated various possible adsorption configurations on Sn(100) and S-doped Sn(100) surfaces for *OCHO, and took the most stable configuration as the active site. For *H, different adsorption sites have also been calculated, and the same sites as *OCHO was chosen to compare adsorption energies.

As suggested by Reviewer #2, we have, in our:

1) R-SI, added **Figure R15** as **Supplementary Fig. 26**.

2) R-MS, p.17, para. 1, included the following clarifying text:

*'Herein, various possible adsorption configurations were considered, and the most stable adsorption structure for *OCHO was taken as the reaction active site (Supplementary Fig. 26).'*

END OF RESPONSE TO REVIEWS

REVIEWERS' COMMENTS

Reviewer #1 (Remarks to the Author):

The reviewer has carefully read the revised manuscript, which indeed has addressed all the concerns in high quality. As thus, I would like to recommend the acceptance of this work in Nature Communications.